# Ultrathin, ultralight dual-scale fibrous networks with high-infrared transmittance for high-performance, comfortable and sustainable $PM_{0.3}$ filter

Yuchen Yang[1,2,3], Xiangshun Li[3], Zhiyong Zhou[3], Qiaohua Qiu[3,4], Wenjing Chen[3], Jianying Huang [1,2], Weilong Cai[1,2], Xiaohong Qin [3] ✉ & Yuekun Lai [1,2] ✉

Highly permeable particulate matter (PM) can carry various bacteria, viruses and toxics and pose a serious threat to public health. Nevertheless, current respirators typically sacrifice their thickness and base weight for high-performance filtration, which inevitably causes wearing discomfort and significant consumption of raw materials. Here, we show a facile yet massive splitting eletrospinning strategy to prepare an ultrathin, ultralight and radiative cooling dual-scale fiber membrane with about 80% infrared transmittance for high-protective, comfortable and sustainable air filter. By tailoring antibacterial surfactant-triggered splitting of charged jets, the dual-scale fibrous filter consisting of continuous nanofibers (44 ± 12 nm) and submicron-fibers (159 ± 32 nm) is formed. It presents ultralow thickness (1.49 μm) and base weight (0.57 g m$^{-2}$) but superior protective performances (about 99.95% $PM_{0.3}$ removal, durable antibacterial ability) and wearing comfort of low air resistance, high heat dissipation and moisture permeability. Moreover, the ultralight filter can save over 97% polymers than commercial N95 respirator, enabling itself to be sustainable and economical. This work paves the way for designing advanced and sustainable protective materials.

As one of the most basic but vital activities of human beings, respiration shoulders the responsibility of transferring mass and energy through the clean air to maintain the normal metabolism of the human body[1]. The suspended particulate matters (PMs) with very small size in the air can carry a variety of pathogenic bacteria, viruses and toxics and thus pose a serious threat to public health via the respiratory system of the individuals[1–3]. Notably, tiny PMs (aerodynamic diameter ≤0.3 μm) are highly permeable and can induce serious respiratory diseases, cardiovascular diseases and even take our lives, which have become another crime culprit to the life safety[1–3]. Considerable facts in recent years have proved the damage of polluted PMs, such as severe acute respiratory syndrome coronavirus (SARS-CoV), influenza virus, Middle East respiratory syndrome (MERS)-CoV, human rhinovirus, and respiratory syncytial virus (RSV)[1,2]. Particularly, the abrupt outbreak of COVID-19 triggered by fatally novel coronavirus have unexpectedly caused billions of infections and tens of millions of deaths and been threatening the living, health and influencing the daily life of all human beings[1,2,4]. Therefore, the protective materials and devices to effectively prevent human body from the harm of polluted air are urgently anticipated.

Fibrous nonwoven filters with the isotropical porous structure can effectively block the airborne pollutants entering the respiratory

[1]Qingyuan Innovation Laboratory, Quanzhou 362801, P. R. China. [2]College of Chemical Engineering, Fuzhou University, Fuzhou 350116, P. R. China. [3]Key Laboratory of Textile Science & Technology of Ministry of Education, College of Textiles, Donghua University, Shanghai 201620, P. R. China. [4]College of Textile Science and Engineering, Zhejiang Sci-Tech University, Hangzhou 310018, P. R. China. ✉e-mail: xhqin@dhu.edu.cn; yklai@fzu.edu.cn

system and is the optimal candidate for body protection[4–9]. The melt-blown nonwovens utilized for commercial face mask always possess the large fiber diameters (>1 μm) and pore size (over 5 μm) and thus show limited $PM_{0.3}$ filtration efficiency (<99%)[10–12]. Besides, they capture $PM_{0.3}$ at the expense of their thickness (>90 μm) and base weight (>20 g m$^{-2}$), inevitably resulting in heat-moisture discomfort and unnecessary consumption of raw materials[10–12]. By contrast, the emerging solution-electrospinning technique brings the dawn of reducing fiber size to less than 1 μm and pore size of filters to 2-5 μm[13,14]. The resultant submicron fibrous mats from a variety of polymers can easily achieve more than 99% filtration efficiency to $PM_{0.3}$[15–18]. Nevertheless, due to the pseudo-nanoscale fiber diameter (>100 nm), the high-performance electrospun mats of over 99.9% $PM_{0.3}$ removal exhibit densely packed structure and thus poor breathing resistance, heat dissipation and moisture diffusion[19–22]. Based on electret, triboelectric and piezoelectric effect, moreover, various charged electrospun submicron fibers have been developed to integrate the high filtration efficiency and the low pressure drop of face mask[23–31]. Unfortunately, above electro-assisted air filters usually possess the serious drawbacks of charge dissipation and nondurable filtration performance, especially in humid environments[32]. Thus, a series of self-powered fibrous filters driven by respiration have been designed to provide its long-term stability of filtration efficiency, but show complicated filter structure which is unfavorable to the wearing comfort[32–35]. Furthermore, tremendous attempts have been done to exploit true nanofiber (<100 nm) filters with gradient structure for balancing filtration efficiency and air resistance[36]. For example, carbon nanotube and supramolecular nanofiber composite filters for $PM_{0.3}$ filtration were respectively fabricated by the spraying-netting and self-assembly strategies[37–41]. Besides, multi-scale nano-/submicron-fiber composite filters, with the hierarchically bead-on-string, nano-net, cage-like, tree-like and other structures, were also exploited[42–54]. As expected, these filters showed efficient $PM_{0.3}$ capture (over 99.9%) and low filtration resistance (less than 200 Pa). Despite above progress, however, the intrinsic conflict between filtration efficiency and wearing comfort of filters has not been satisfactorily addressed. And the preparation of ultrathin, ultralight fibrous mats of simultaneous high-performance filtration, wearing comfort and sustainability remains to be highly challenging.

Herein, a facile and massive one-step strategy was proposed to controllably prepare the high-protective, comfortable and sustainable filter. Through tailoring the antibacterial surfactant-induced splitting of charged jets, the hierarchically dual-scale fibrous filter with high infrared transmittance of more than 80% is prepared by the double-needle electrospinning technology. Benefiting from the very fine fibrous size (nanofibers of $44 \pm 12$ nm, submicron-fibers of $159 \pm 32$ nm) together with the micro-gradient architecture, the membrane presents superior protective performances of over 99.95% $PM_{0.3}$ removal and durable antibacterial activity, even at ultralow thickness of 1.49 μm and base weight of 0.57 g m$^{-2}$. Simultaneously, the ultrathin filter also shows high visible light transmittance of 84.5% and prominent wearing comfort of low air resistance (120 Pa), high heat dissipation (4–9 °C lower than commercial face mask) and moisture permeability (7541 g m$^{-2}$ day$^{-1}$). Notably, in comparison to the commercial melt-blown filters (over 20 g m$^{-2}$), the ultralight filter with the base weight of 0.57 g m$^{-2}$ can save more than 97% raw materials, which is proved to be more sustainable and economic.

## Results

### Design and preparation of dual-scale fibrous filter

The design of the high-protective, comfortable and sustainable air filter was obeyed to the following criteria: i) the filter can simultaneously filtrate the highly permeable $PM_{0.3}$ and kill the harmful bacteria or virus, to avoid their sustained reproduction and transmission, ii) the filter must be ultrathin and infrared transparent and have the

hierarchical structure to enable itself of excellent heat dissipation, moisture permeability and low breathing resistance, and iii) the filter should possess extremely low base weight for reducing the consumption of raw materials and thus promoting the sustainability of disposable face mask. To satisfy above three requirements, an anti-bacterial ionic surfactant-induced polyamide 6 (PA6) dual-scale fibrous networks, composed of nanofibers ($44 \pm 12$ nm) and submicron-fibers (about $159 \pm 32$ nm), were created (Fig. 1a). The hierarchically nano-/submicron-fibrous structure was designed to endow the filter with the capacity of high-efficiency $PM_{0.3}$ filtration and low filtration resistance at ultralight and ultrathin state, which also contributed to the superior heat and moisture transfer and sustainability of face mask.

Figure 1b displays a piece of dual-scale fiber filter with a dimension of 0.3 m by 0.7 m. As shown in Fig. 1c, the massively prepared membrane showed the ultralow thickness of only 1.49 μm (55 times thinner than single human hair of 82 μm). Figure 1d exhibits that the weight of a piece of dual-scale fiber filter was only 0.0124 g, which was 47 times lighter than the filter layer of commercial N95 mask (Supplementary Fig. 1). The breathable filter also allowed the ammonia to go through and discolor the pH test paper within 1 s (Fig. 1d). Moreover, due to the structural and infrared transparent characteristics, the dual-scale fibrous filter presented higher surface temperature of about 28.0 °C than N95 mask of about 19.3 °C, showing better heat dissipation (Fig. 1f).

To prepare the comfortably high-performance air filter, a facile but effective strategy was proposed to form a hierarchical structure of nano-/submicron-fibers by the electrospinning technology. During the electrospinning process, the spinning solution near the needle tip would firstly form into the "Taylor cone" and then evolve into stretched jets consisting of straight trajectory and whipping-triggered spiral trajectory under the effect of high-voltage electrostatic field (Fig. 2a)[55]. Afterwards, the submicron-fibers could be prepared by the synergistic effect of high drafting of polymer jets and solvent volatilization (Fig. 2a)[55]. Considering that the properties of precursor solution strongly influence the formation of fibers, the amphiphilic dodecyl-trimethylammonium chloride (DTAC), composed of hydrophilic ammonium group and lipophilic alkane chain, was introduced to simultaneously tailor the conductivity, viscosity and surface tension of PA6 solution by ion-dipole interaction[44,46,50]. Under the effect of electrostatic field, DTAC could be assembled around PA6 molecular chains and significantly weaken molecular chain interaction and promote their electrostatic repulsion, which beneficially resulted in the splitting of dilute jets and the formation of dual-scale fibers[44,46,50]. Moreover, the hydrophilic DTAC of jet surface could also attract the bad solvent (water) to replace the good solvent (Hexafluoroisopropanol, HFIP) and solidify dual-scale fibers. The amphiphilic DTAC could easily dissolve and disperse in the PA6 solutions (Fig. 2a). Figure 2b presents the effect of DTAC on solution properties. Different from the pure PA6 solutions, 5.5 wt% PA6 solutions doped with different DTAC ratio presented the enhanced conductivity and the decreased surface tension and viscosity. As the doped content of DTAC increased from 0 wt% to 12 wt%, the conductivity of PA6/DTAC solutions raised from 3 to 454 μS/cm while the viscosity of PA6/DTAC solutions decreased from 166 to 138 mPa·S, and the surface tension of PA6/DTAC solutions reduced from 20.53 to 17.80 mN/m.

Notably, electrospun PA6 fibers produced from dilute precursor solution showed the beaded structure (Supplementary Fig. 2a–c). But the beads within the PA6 submicron-fibers gradually decreased with the increased solution concentration from 3 to 5 wt%, and they completely disappeared when the solution concentration exceeded 5.5 wt% (Fig. 2c and Supplementary Fig. 2d). The formation of eletrospun beaded fibers results from the Rayleigh instability of polymeric jets[13]. Dilute jets from low viscosity solution exhibits poor chain entanglement, which resulted in their uneven stretching (Rayleigh instability)

and formed into contracted beads[13]. The increased solution concentration enhanced its chain entanglement and viscosity, which enabled jets to effectively resist the Rayleigh instability and form into non-beaded fibers[13]. As expected, the dual-scale fibrous networks emerged by electrospinning DTAC-doped 5.5 wt% PA6 solutions (Figs. 2d-g and Supplementary Fig. 3). The finer nanofibers within networks increased first and then decreased when the content of DTAC increased from 1, 3, and 6 to 12 wt% and were the most in PA6/DTAC-2 mats and were the least and broken in PA6/DTAC-4 mats (Fig. 2d-g). As shown in Fig. 2h-i, the average diameter of submicron-fibers produced from 5.5 wt% PA6 solution is 179 nm while the dual-scale fibrous mat in PA6/DTAC-2 consists of true nanofibers and submicron-fibers with the mean diameter of 44 nm and 159 nm. Interestingly, the diameter sum of nanofibers and submicron-fibers in dual-scale fibrous mat closely approximated to the size of pure PA6 submicron-fibers, which further demonstrated that the DTAC-triggered jet splitting resulted in the formation of dual-scale fibrous networks. The prepared true nanofibers with the mean diameter of 44 nm were much thinner than those fibers produced by solution or melt electrospinning, solution or melt blow spinning techniques (Fig. 2j).

## Characteristics of dual-scale fibrous networks

The structural and physical characteristics of DTAC modified PA6 fiber mats were systematically investigated. As shown in Fig. 3a, the

wettability of electrospun PA6 submicron-fiber mats were significantly affected by the amount of doped DTAC. The pure PA6 submicron-fiber mats were slightly hydrophobic and showed stable water contact angle (WCA) of 118° after 20 seconds. Differently, PA6/DTAC-based mats exhibited enhanced hydrophilic with the increased DTAC contents but could not be fully wetted due to the limited membrane thickness, which may enable the prepared filter of high filtration efficiency even at the humid environment.

Besides, in comparison to the concentrated pore size of electrospun PA6 submicron-fiber mat (1.634 µm), PA6/DTAC-1, PA6/DTAC-2, PA6/DTAC-3 and PA6/DTAC-4 fiber membranes showed decreased pore size of 1.326 µm, 0.933 µm, 0.938 µm and 0.967 µm, respectively (Fig. 3b). Notably, the PA6/DTAC-2 mats presented the smallest pores due to its dual-scale fibrous structure consisting of true nanofibers. And the pore size of PA6/DTAC-2 fiber mat decreased with the increase of spinning time and was gradually constant when the spinning time exceeded 50 min (Fig. 3c). Furthermore, as shown in Fig. 3d, the calculated porosity of as-prepared fiber mats increased from 41% to over 65% as the increase of DTAC concentration.

Figure 3e presents the base weight of various PA6 and PA6/DTAC fiber mats with different spinning time. Obviously, the base weight of all fiber mats was positively correlated with the electrospinning time. Pure PA6 submicron-fiber mat of electrospinning 60 min possessed the highest basic weight of 1.81 g m$^{-2}$. By contrast, the DTAC-doped fiber mats showed the significantly decreased base weight and their

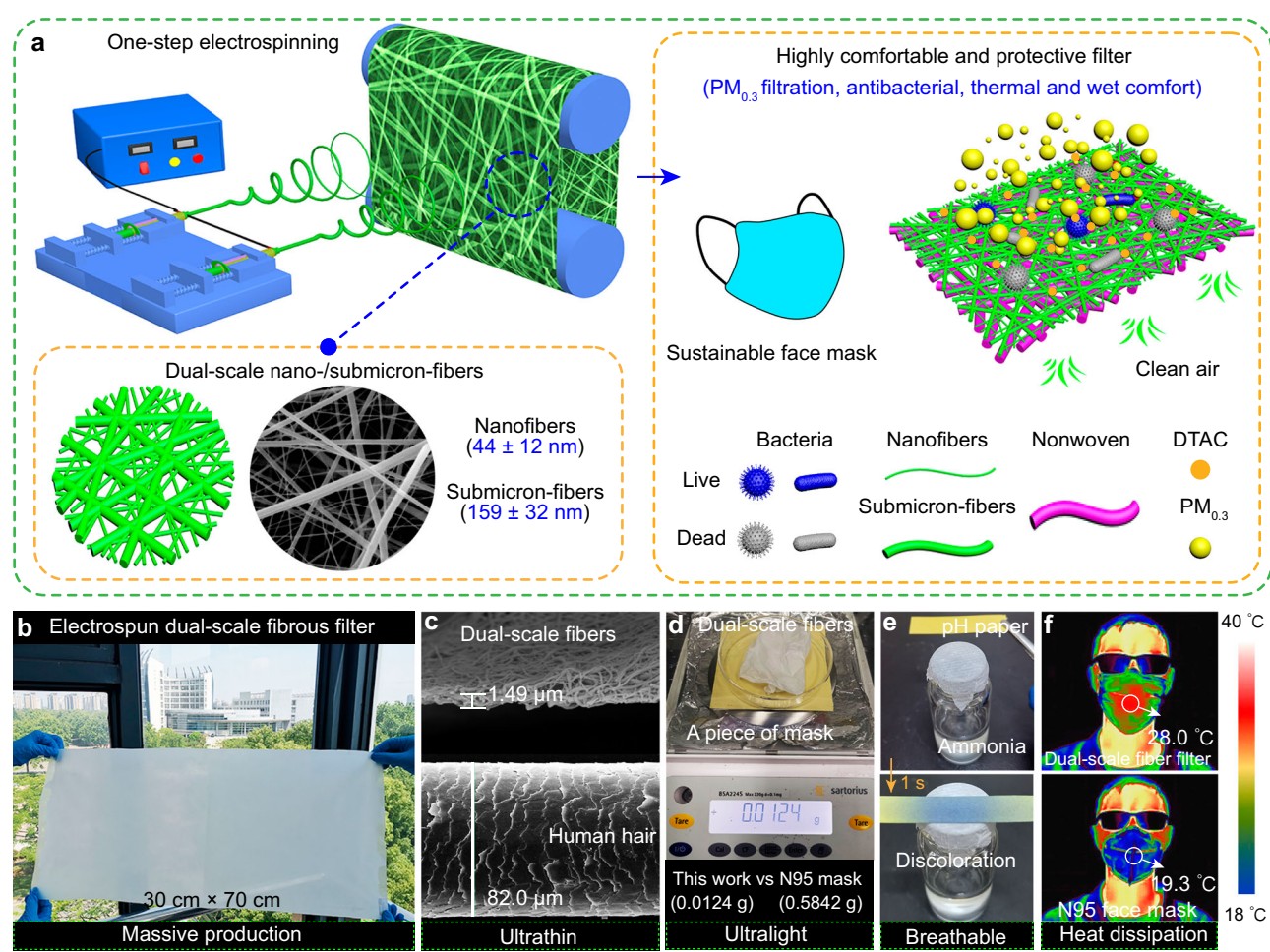

**Fig. 1 | Ultrathin, ultralight dual-scale fibrous filter with high protective capacity and wearing comfort. a** Preparation process and structural characteristics of dual-scale fibrous filter. **b** Optical image of the dual-scale fiber filter. **c** Scanning electron microscope (SEM) images of the dual-scale fibrous filter and human hair. **d** Weight of a piece of dual-scale fibrous mask. **e** The breathability of the dual-scale fibrous filter. **f** Comparison of heat dissipation of the dual-scale fibrous filter and N95 mask.

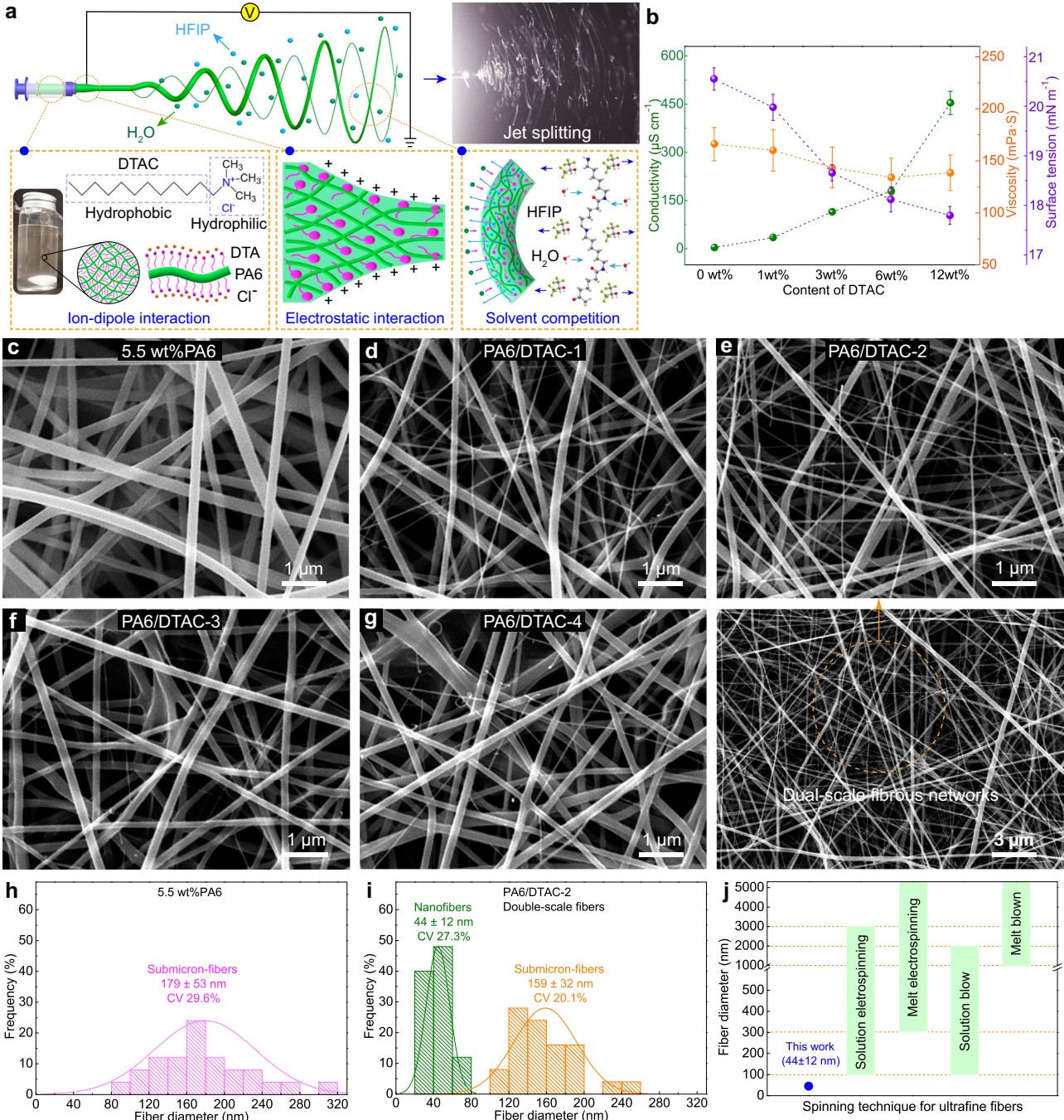

**Fig. 2 | Formation of electrospun dual-scale fibrous networks. a** Diagram and optical image of the formation of dual-scale fibers. **b** Properties of 5.5 wt% PA6 solutions doped with various DTAC contents. SEM images of (**c**) PA6, **d** PA6/DTAC-1, **e** PA6/DTAC-2, **f** PA6/DTAC-3, **g** PA6/DTAC-4 fiber filters. Diameter distribution of (**h**) 5.5 wt% PA6 fibers and (**i**) dual-scale fibers. **j** Fiber diameter comparison of this work and other spinning techniques. Error bars represent the standard deviation of the measured properties and the samples are at least three. Source data are provided as a Source Data file.

base weight was negatively correlated with the doped DTAC content. The mean base weight of PA6/DTAC-2 fiber mats with the electrospinning time of 10, 20, 30, 40, 50, 60 min was separately 0.13, 0.24, 0.36, 0.45, 0.57, 0.69 g m$^{-2}$. Similarly, PA6 submicron fiber mats showed the maximum thickness of 2.22 μm while the thickness of PA6/DTAC-1, PA6/DTAC-2, PA6/DTAC-3 and PA6/DTAC-4 mats was separately 1.63, 1.49, 1.08 and 0.908 μm (Fig. 3f). The core filtration layer of two kinds of commercial face masks including medical mask (MM) and melt-blown mask (MBM, N95) respectively possessed the high base weight of 22.9 and 25.6 g m$^{-2}$ and the high thickness of 91.2 and 129 μm, which usually resulted in their wearing discomfort and unsustainability

(Fig. 3g). Differently, the prepared dual-scale fibrous networks (PA6/DTAC-2) exhibited more than 40 times lower base weight and 60 times thinner thickness (Fig. 3g).

Due to the decreased thickness and base weight, as shown in Fig. 3h, i, the visible light transmittance of electrospun PA6 submicron fiber mat (69.2%) was higher than MM and MBM, but much lower than DTAC-doped fibrous mats. And PA6/DTAC-4 displayed the highest visible light transmittance of 89.1% because of its lowest thickness of 0.908 μm while that of PA6/DTAC-2 fiber mat is 84.5%. In addition, the optical images of filters shown in Fig. 3j also coincided with the results of Fig. 3f. The ultralight and ultrathin characteristics of this mat are

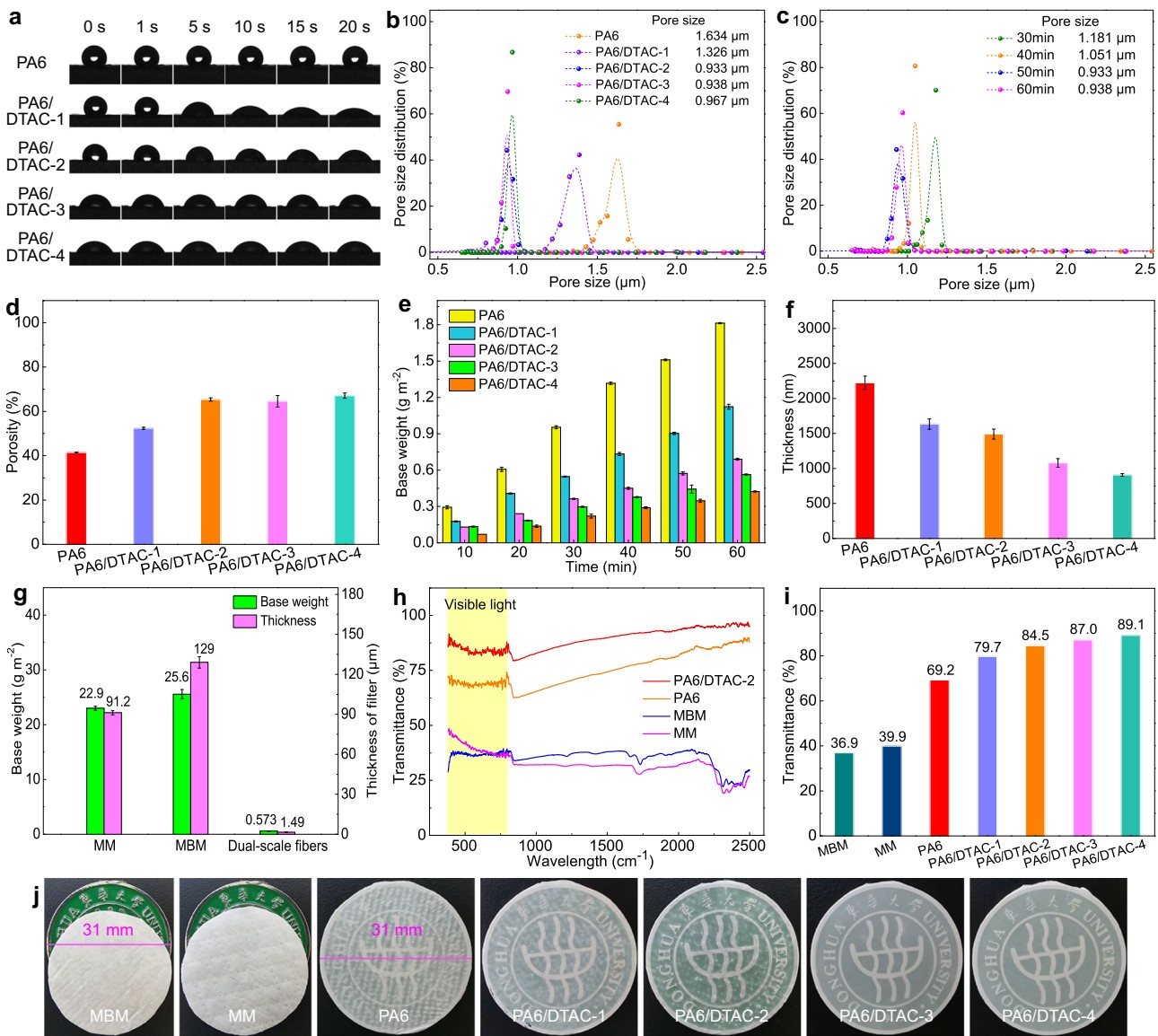

**Fig. 3 | Physcial characteristics of dual-scale fibrous membranes. a** Wettability of various PA6 and PA6/DTAC fiber mats. **b** Pore size distribution of various fiber mats. **c** Pore size distribution of PA6/DTAC-2 dual-scale fibrous mats with different electrospinning time. **d** Porosity of various fiber mats. **e** Base weight of various fiber mats with different electrospinning time. **f** Thickness of various fiber mats. **g** Base weight and thickness comparison of MM, MBM and PA6/DTAC-2 dual-scale fiber filters. **h** Total transmittance curves of various fiber mats. **i** Visible light transmittance comparison of MBM, MM, PA6 and various PA6/DTAC fiber mats. **j** Optical images of various fiber mats. Error bars represent the standard deviation of the measured properties and the samples are samples are at least three. Source data are provided as a Source Data file.

anticipated to effectively enhance the wearing comfort of filter and simultaneously saving raw materials.

## Filtration performance of dual-scale fibrous filters

The overall $PM_{0.3}$ filtration performance of the obtained filters was systematically investigated. After filtration, the sodium chloride (NaCl) aerosols with the average size of below 260 nm ($PM_{0.3}$) were partly captured by PA6 submicron-fibers (Fig. 4a). Differently, the dual-scale fibrous networks (PA6/DTAC-2) consisted of true nanofibers and submicron-fibers and thus showed the significantly improved capture ability of the highly permeable $PM_{0.3}$ (Fig. 4b, c). From Fig. 4c, obviously, thinner nanofibers whose diameter was below 100 nm had stronger $PM_{0.3}$ capture capability and thicker submicron-fibers could also filtrate part of $PM_{0.3}$. Figure 4d and Supplementary Fig. 4 show the relationship between filtration properties and electrospinning time of PA6 and PA6/DTAC fiber filters. It indicated that both filtration

efficiency and filtration resistance of PA6 fiber and DTAC-doped PA6 fiber filters increased with the increase of electrospinning time from 10 min to 60 min, which resulted from the synchronously increased base weight and decreased pore size of mats. Notably, the doped DTAC effectively reduced the diameter of PA6 fibers, which endowed PA6/DTAC fiber filters with superior filtration efficiency and quality factor (QF). For instance, the PA6 submicron-fiber filter (spinning time: 60 min, base weight: 1.8 g m$^{-2}$) showed the $PM_{0.3}$ capture efficiency of 98.8% and pressure drop of 129 Pa. The PA6/DTAC fiber filters (base weight <1.0 g m$^{-2}$) possessed over 99.9% $PM_{0.3}$ filtration efficiency, which fully embodied the filtration advantage of true nanofibers (below 100 nm) to the highly permeable particles. Among all kinds of PA6/DTAC fiber mats, moreover, the PA6/DTAC-2 filters own the optimal $PM_{0.3}$ filtration performance due to its more ideal dual-scale fibrous structure. In details, they exhibited the high efficiency and low air resistance of 99.5% and 72 Pa (spinning time: 30 min, base weight:

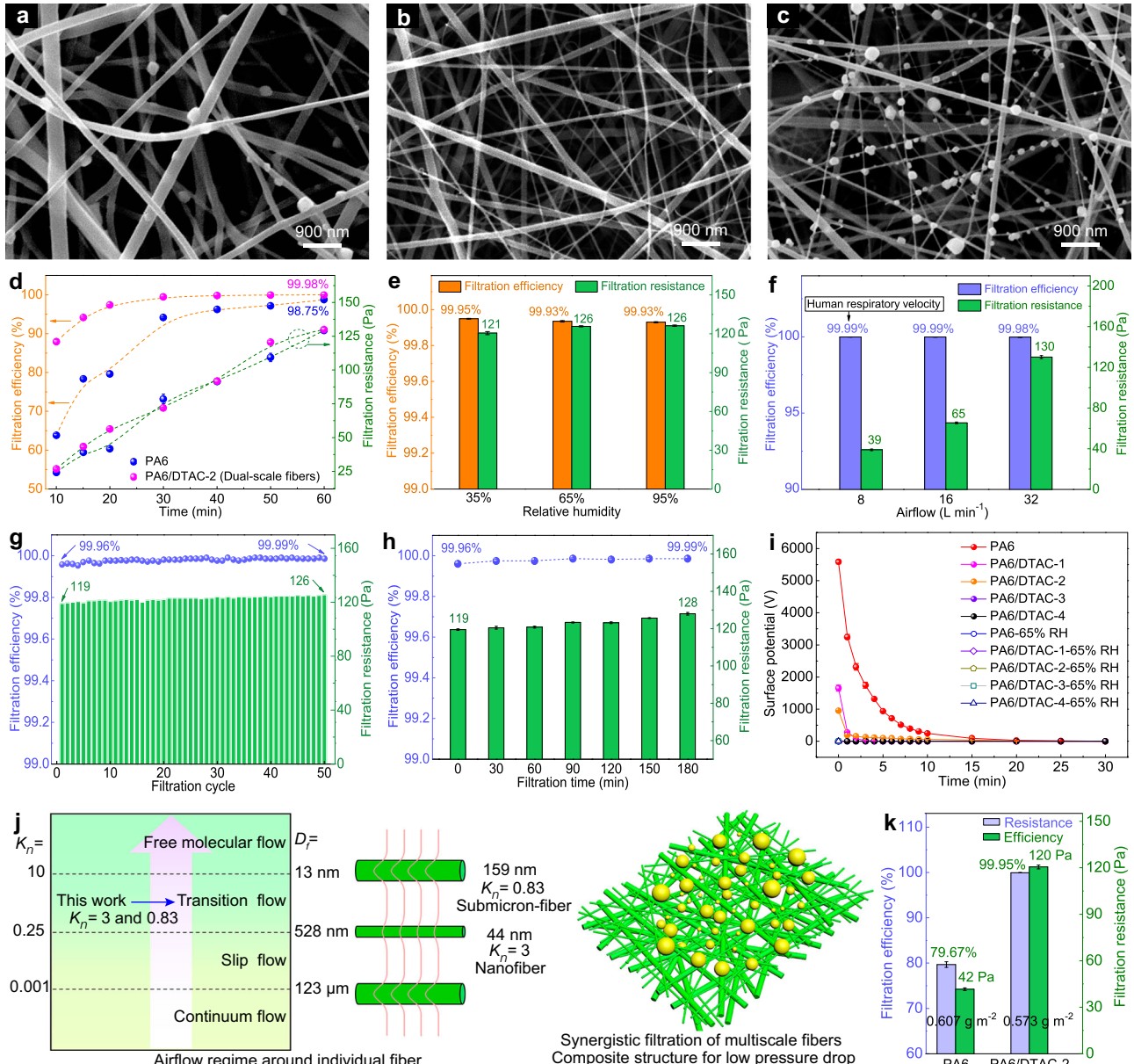

**Fig. 4 | Filtration properties and mechanism of dual-scale fibrous filters.** SEM images of (**a**) the PA6 submicron-fiber filter after filtration and the PA6/DTAC-2 filter (**b**) before filtration and (**c**) after filtration. **d** Filtration properties of PA6 and PA6/DTAC-2 filters. Filtration performances of PA6/DTAC-2 filter (**e**) placed in various relative humidity and (**f**) under different airflow velocity. **g** Filtration cycle and (**h**) long-term filtration of the dual-scale fibrous filter. **i** Surface potential attenuation of PA6 and various PA6/DTAC filters with and without humidifying treatment. **j** Airflow regime around individual fibers with various $K_n$ and size and filtration diagram of the PA6/DTAC-2 filter. **k** Filtration performance comparison between PA6 and PA6/DTAC-2 filter with approximate base weight. Error bars represent the standard deviation of the measured properties and the samples are samples are at least three. Source data are provided as a Source Data file.

0.36 g m$^{-2}$), 99.95% and 120 Pa (spinning time: 50 min, base weight: 0.57 g m$^{-2}$), 99.98% and 130 Pa (spinning time: 60 min, base weight: 0.69 g m$^{-2}$). Meantime, the PA6/DTAC-2 fiber networks could also achieve the highest QF of 0.072 when the PM$_{0.3}$ filtration efficiency of all filters was above 99%.

Considering the variability of face mask application scenarios, the stability of filtration property of PA6/DTAC-2 filter continuously placed at different humidity environments for 24 h has also been evaluated, as shown in Fig. 4e. The PA6/DTAC-2 filter presented almost unchanged filtration efficiency but increased pressure drops about 5 Pa when the treated ambient humidity increased from 35 ± 5% to 65 ± 5%. Differently, its filtration efficiency and resistance did not change when the treated ambient humidity increased from 65 ± 5% to 95 ± 5%. The increased air resistance may derive from the fact that moisture

absorption of hydrophilic PA6/DTAC-2 fibers caused the slight decrease of fiber packing density. And when hydrophilic PA6/DTAC-2 filter achieved moisture absorption balance at 65 ± 5% relative humidity, its breathing resistance did not rise further even at the ultrahigh relative humidity of 95 ± 5%. Above results strongly clarified the stable filtration performance of dual-scale fibrous filter under different humidity environments. The correlation between filtration performance of filter and airflow velocity was also studied, as shown in Fig. 4f. When the airflow velocity decreased from 32 to 16 L min$^{-1}$, the PM$_{0.3}$ filtration efficiency and pressure drop of the PA6/DTAC-2 filter increased from 99.95% and 120 Pa to 99.99% and 65 Pa, respectively. It even showed 99.99% PM$_{0.3}$ capture efficiency and ultralow air resistance of only 39 Pa at the airflow velocity of 8 L min$^{-1}$ (the respiratory velocity of common human). Besides, the service life of the prepared

filter has been tested as shown in Figs. 4g and 4h. After 50 filtration cycles, its $PM_{0.3}$ filtration efficiency and filtration resistance respectively increased from 99.96% to 99.99% and 119 Pa to 126 Pa (Fig. 4g). Similarly, after 3 hours of long-term filtration test, the $PM_{0.3}$ filtration efficiency and filtration resistance of the dual-scale fibrous filter also increased from 99.96% to 99.99% and 119 Pa to 128 Pa (Fig. 4h). The results indicated that the designed filter showed superior filtration durability and service life. The increase in both filtration efficiency and pressure drop was mainly due to pore clogging caused by the accumulation of specific substances in porous fibrous filter after long-term filtration[48].

To clarify the filtration mechanism of the high-performance dual-scale fibrous filter, as shown in Fig. 4i, the surface potential of PA6 and various PA6/DTAC fibrous filters was measured. The electrospinning technique can achieve the in situ charging of as-prepared polymer fibers under high-voltage electrostatic field[25,48,49]. Thus, pure PA6 submicron-fiber mat showed ultrahigh surface potential of over 5500 V. However, the modified PA6 fibrous filters displayed significantly decreased surface potential as the increase of DTAC content, and the surface potential of PA6/DTAC-4 filter was even less than 20 V. Supplementary Fig. 5 also shows that PA6 submicron-fiber mat could attract lots of foam balls via electrostatic interaction and PA6/DTAC-2 filter could not. Furthermore, all above filters showed the significant potential decay under the test conditions of 50% relative humidity (RH) and 25 °C. The surface potential of PA6 submicron-fiber filter had longer decay time of over 15 min while the decay time of PA6/DTAC-1 and PA6/DTAC-2 filters did not exceed 5 min. The phenomenon could be attributed to the differences in surface properties of PA6 submicron-fibers and PA6/DTAC composite fibers. In contrast to pure PA6 fibers, PA6/DTAC composite fibers possessed superior hydrophilic ability and charge neutralization capability due to the quaternary ammonium ion of DTAC, which enabled themselves of fast dissipating their surface charges under the humidity environment. When continuously placed at the atmosphere of 65% RH for 24 h, the surface potential of all filters attenuated to almost 0 V. Above results strongly demonstrated that the PA6/DTAC-2 dual-scale fibrous filter did not capture the highly permeable $PM_{0.3}$ by electrostatic force. It is well-known that the smaller the particle size, the more significant the brownian motion. From the Figs. 4a and 4c, the NaCl aerosols displayed much smaller size (below 300 nm) and thus excellent diffusion ability, which created the prerequisite for porous filter to efficiently capture these $PM_{0.3}$. Therefore, the dual-scale fibrous filter consisting of nanofibers and submicron-fibers had very small pore size of less than 1 μm, which made it possible to effectively filtrate the easily diffusible $PM_{0.3}$ by the synergistic effect of physical interception and brownian diffusion (Supplementary Fig. 6)[40,41,50].

Additionally, the prepared PA6/DTAC-2 filter also showed low pressure drop, which resulted from its ultrathin thickness and ultralight base weight and submicron-/nano-fiber composite structure. To reveal the reason, the Knudsen number ($K_n$) was utilized to reflect the airflow regime around individual fibers during the filtration (Fig. 4j). According to the $K_n$, the airflow modes around single fibers could be divided into four types: continuum flow ($K_n < 0.001$, fiber diameter ($D_f$) > 132 μm), slip flow ($0.001 < K_n < 0.25$, 528 nm $< D_f < 132$ μm), transition flow ($0.25 < K_n < 10$, 13 nm $< D_f < 528$ nm) and free molecular flow ($K_n > 10$, $D_f < 13$ nm). The PA6/DTAC-2 dual-scale fibrous filter was composed of true nanofibers with the $K_n$ of 3 and submicron-fibers with the $K_n$ of 0.83. During the filtration process, the airflow through the dual-scale fibrous networks was at the transition flow regime and would be subjected to the strong drag effect from nanofibers. Fortunately, submicron-fibers with much lower $K_n$ in the composite structure filter can weaken the packing density of nanofibers and thus effectively enhance the air permeability[37]. More importantly, due to the super-strong capture ability of dual-scale fibers to $PM_{0.3}$, the prepared PA6/DTAC-2 filter

could filtrate 99.95% $PM_{0.3}$ even at the ultrathin thickness of only 1.49 μm and ultralight base weight of only 0.57 g m$^{-2}$ (Fig. 4k). As a result, the ultrathin and ultralight characteristics of the filter endowed itself with the low pressure drop of 120 Pa. In contrast, PA6 submicron-fiber mat with the base weight of 0.60 g m$^{-2}$ showed much lower $PM_{0.3}$ filtration efficiency of 79.67% and pressure drop of 42 Pa because of thicker fiber diameter and higher $K_n$.

## Wearing comfort and antibacterial property of the filter

Wearing comfort is another critical aspect for protective face mask but usually neglected. As shown in Fig. 5a, the heat and moisture from the human body need to be dissipated into the atmosphere to maintain physiological comfort of the individuals. However, the existing core filter materials usually achieve over 99.9% $PM_{0.3}$ removal at the expense of thickness and base weight, which seriously hinders the transfer of heat and moisture from the skin to the external environment. To resolve the problem, the ultrathin and ultralight dual-scale fibrous network composed of true nanofibers and submicron-fibers was designed to simultaneously filtrate $PM_{0.3}$ and promote the heat and moisture dissipation. Ultrathin and ultralight porous structure could enhance the thermal conduction and convection. Notably, PA6 polymer utilized as the raw materials for electrospinning was turned out to be infrared transparency while submicron-scale and nano-scale fiber size also determined the poor molecular vibration and low infrared emissivity[56]. High-performance protective filter referred to the one that can both effectively intercept the highly permeable particles and kill the pathogenic bacteria carried by the suspended PM. As a cationic surfactant, particularly, the DTAC could not only tailor the precursor solution properties for producing dual-scale fibers, but also be antibacterial by release-killing or contact-killing[33]. Hence, the designed PA6/DTAC-2 dual-scale fibrous filter was desired to concurrently possess high protective performance ($PM_{0.3}$ removal and antibacterial property) and high comfort (moisture transmission and thermal dissipation).

To verify above design concept, the wearing comfort and antibacterial property of PA6/DTAC-2 filter were systematically evaluated. Figure 5b shows that the water vapor transmission of PA6/DTAC-2 filter (7541 g m$^{-2}$ day$^{-1}$) was, respectively, 1.12 and 1.17 times higher than that of commercial MM and MBM on account of its ultrathin and porous structure. Nevertheless, the air permeability of it was only 96 mm/s under the pressure difference of 200 Pa, which was separately 38.1% and 43.4% of that of MM and MBM (Fig. 5c). The result mainly derived from the submicron pore-induced strong drag effect under high pressure difference. In the Fig. 5d, the yellow area represented that the human body emitted mid-IR wavelengths of 3 to 16 μm, resulting in the most of heat loss (over 50%)[56]. As expected, the PA6/DTAC-2 filter held superior infrared transmittance of about 80% at the wavelengths of 3 to 16 μm, which was much higher than that of MM (about 40%) and MBM (about 5%). It indicated that the designed dual-scale fibrous mat owned excellent radiative cooling effect by transmitting the human body radiation. To further illustrate the heat dissipation of the PA6/DTAC-2 filter-based face mask, the temperature variations on its outer surface during the inhalation and exhalation cycles was recorded in real-time by infrared camera, as presented in Fig. 5e. The outer surface temperature of face mask increased from 22.6 to 27.7 °C and then returned to nearly 22.6 °C during the body respiratory cycle, showing great cooling ability of about 5.1 °C. In comparison, the commercial MM, MBM, KN95 mask exhibited inferior cooling performance of about 4.1, 1.1 and 0.9 °C, respectively (Fig. 5f). Moreover, the surface temperature of PA6/DTAC-2 filter during the inhale process was also higher than others. These results verify that the PA6/DTAC-2 dual-scale fibrous filter can be utilized as the air-conditioning face mask for promoting the thermal comfort of wearers.

Figure 5g presents the growth of *Staphylococcus aureus* (*S. aureus*) and *Escherichia coli* (*E. coli*). In the blank sample (with the PA6

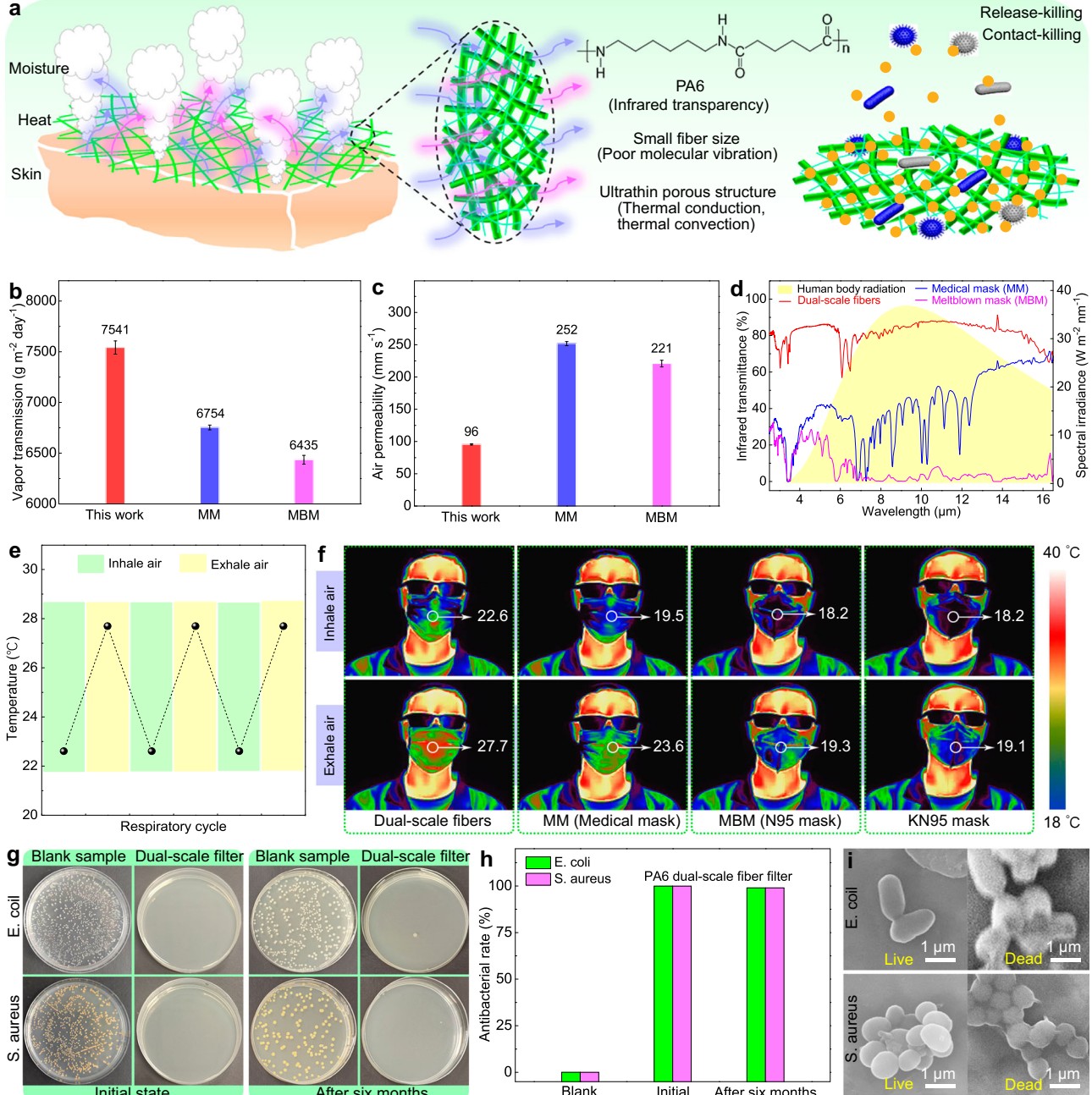

**Fig. 5 | Wearing comfort and antibacterial perfromances of dual-scale fibrous filters. a** Diagrams of heat and moisture transfer of dual-scale fibrous filter and its antibacterial mechanism. **b** Water vapor transmission, (**c**) air permeability and (**d**) infrared transmittance of dual-scale fiber filter, MM and MBM. **e** Surface temperature of face mask during the respiratory cycle. **f** IR images of dual-scale fiber filter, MM, MBM, KN95 mask. **g** Quantitative antibacterial images and (**h**) antibacterial rate of blank sample and dual-scale fiber filter. **i** SEM images of live and dead bacteria. Error bars represent the standard deviation of the measured properties and the samples are samples are at least three. Source data are provided as a Source Data file.

filter), both *S. aureus* and *E. coli* grew well and were all over the Petri dishes. Differently, All *S. aureus* and *E. coli* were entirely killed when cultured with the PA6/DTAC-2 filter. Particularly, the PA6/DTAC-2 filter also exhibited excellent antibacterial activities and could killed all bacteria after six months of storage, indicating itself the durably antibacterial properties. As shown in Fig. 5h, the PA6 filter (blank sample) showed no antibacterial properties while PA6/DTAC-2 dual-scale fibrous filter presented 99.9% inhibition rate for both *E. coli* and *S. aureus* and still more than 99% antibacterial rate after six months of storage. The efficient and long-term antibacterial properties of resultant filter could be attributed to the nano- and submicron-scale

fiber size and sustained release of DTAC from fiber inside to surface[33,57].

To further reveal the bactericidal mechanism of PA6/DTAC-2 filter, the morphological changes of two kinds of bacteria were also investigated (Fig. 5i). The live *E. coli* owned smooth rod-like shape while the live *S. aureus* showed smooth spherical morphology. In sharp contrast, both the surface of died *E. coli* and *S. aureus* deformed and collapsed due to exposure to the antibacterial PA6/DTAC-2 filter. Moreover, some substances flowed from the surface of bacteria and resulted in the severe adhesion between each other. It strongly supported that positively charged DTAC could adsorb the negatively

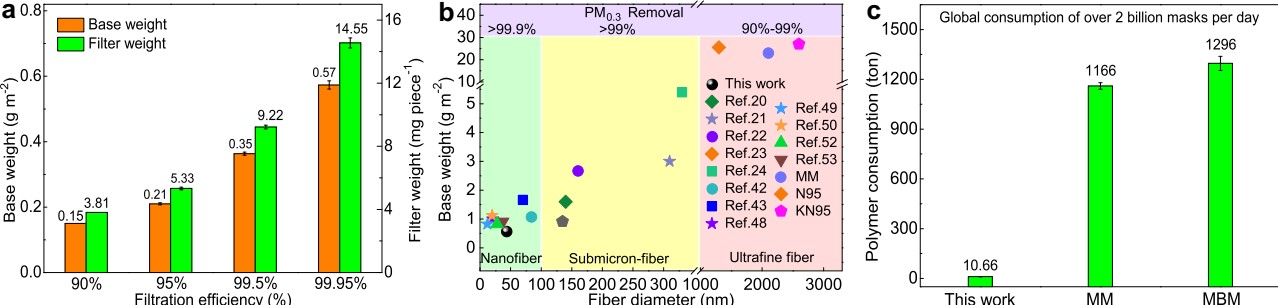

**Fig. 6 | Sustainability of dual-scale fibrous filters. a** Relationships between filtration efficiency and base weight of filter, weight of a piece of filter. **b** Comparison of filtration efficiency, base weight and fiber diameter of dual-scale fiber filter and recent works. **c** Polymer consumption of dual-scale fiber filter, MM and MBM for two billion masks. Error bars represent the standard deviation of the measured properties and the samples are samples are at least three. Source data are provided as a Source Data file.

charged bacteria and thus destroy its structure via electrostatic interaction[33].

## Sustainability of the dual-scale fibrous filter

Due to the nano-scale fiber size and micro-gradient structure, the PA6/DTAC-2 dual-scale fibrous mat owned excellent filtration performances even under the ultralight base weight. Figure 6a shows correlations between weight of the dual-scale fibrous filter and its filtration efficiency. The filter with the ultralow base weight of only 0.15, 0.21, 0.35 and 0.57 g m$^{-2}$ could filtrate 90%, 95%, 99.5% 99.95% PM$_{0.3}$, respectively. And the dual-scale fibrous membrane for preparing a piece of face mask with 90%, 95%, 99.5% and 99.95% PM$_{0.3}$ removal was only 3.81, 5.33, 9.22 and 14.55 mg, separately. Moreover, our designed dual-scale fibrous networks simultaneously possessed ultrafine fiber diameter (44 nm), superior filtration efficiency (about 99.95% PM$_{0.3}$ removal), ultralight base weight (0.57 g m$^{-2}$), showing the high-performance and sustainable characteristics and thus enable itself to be competitive when compared to recent excellent works (Fig. 6b and Supplementary Table 1). Particularly, it is well-known that the global daily consumption of protective face masks exceeded two billion during the COVID-19[11]. Two billion face masks would consume 10.66, 1166 and 1296 tons of polymers when utilizing dual-scale fibrous filter, MM and MBM as the core filtration materials, respectively (Fig. 6c). It illustrated that the designed ultralight filter could sustainably and economically save over 97% raw materials on the premise of ensuring higher protective performance and wearing comfort.

## Discussion

In summary, we proposed a facile and massive methodology to controllably fabricate the high-performance, comfortable, anti-bacterial and sustainable filter with high infrared transmittance of more than 80% via the one-step electrospinning technique. The simple regulation of antibacterial surfactant could trigger the splitting of charged jets and result in the formation of dual-scale fibrous networks, which consisted of continuous nanofibers (44 ± 12 nm) and submicron-fibers (over 159 ± 32 nm). Due to our structure design, the hierarchically dual-scale fiber filter, with the ultrathin thickness of 1.49 μm and ultralow base weight of 0.57 g m$^{-2}$, still exhibited excellent protective properties of over 99.95% PM$_{0.3}$ removal and durable antibacterial activity. Benefiting from its ultrathin, ultralight and infrared transparent characteristics, the filter simultaneously owned high visible light transmittance of 84.5%, great wearing comfort of low air resistance (120 Pa), superior heat dissipation (4–9 °C lower than commercial face mask) and water vapor transmittance (7541 g/m$^{-2}$ day$^{-1}$). Moreover, contrast to the commercial melt-blown filters (over 20 g m$^{-2}$), the ultralight filter (0.57 g m$^{-2}$ for 99.95% PM$_{0.3}$ removal) can save more than 97% raw materials. This work may

provide insight to design advanced nanofiber materials for broad applications in personal protection, energy, environment and others.

## Methods

### Materials

PA6, DTAC (99%), HFIP (99.5%) were bought from Shanghai MacLean Biochemical Technology Co., Ltd.. Polypropylene (PP) nonwoven fabric with an ultralow filtration efficiency (1%) and pressure drop (4 Pa) under an airflow velocity of 32 L min$^{-1}$ was purchased from Nantong Yipinju Fabric Co., Ltd., China.

### Preparation of electrospun fiber mats

The electrospinning solutions were obtained by dissolving PA6 or PA6 and DTAC with different doping ratios into the solvent of HFIP for 12 h under magnetic stirring at ambient temperature. And the details were summarized in Supplementary Table 2. Then, various PA6 and PA6/DTAC fibrous mats were prepared by utilizing a multi-needle electrospinning machine (NEU, DAIEI KAGAKU SEIKI MFG. Co., Ltd., Japan). The speed of collector and reciprocating spinnerets was set to 5 m/min and 10 cm/min, respectively. The fibrous filter with the size of 30×70 cm$^2$ could be uniformly fabricated through rotary collector and reciprocating spinnerets.

### Characterizations

The solution properties (conductivity, viscosity, and surface tension) were measured by conductivity meter (FE32-Standard, Mettler-Toledo Group, Switzerland), viscometer (DV3T, Brookfield Ltd., America), and surface tension meter (DCAT11, Dataphysics Instrument Ltd., Germany), respectively. Jet splitting image was captured via high-speed camera (i-SPEED 716, Nikon, Japan). The morphologies of the electrospun fibrous mats were characterized by scanning electron microscopy (SEM; DXS-10ACKT, Hitachi Group, Japan). The diameter of the electrospun fibers was determined utilizing the Nano Measurer software. The wettability of samples was assessed by the contact angle goniometer (OCA15EC Dataphysics, Germany). The pore sizes and pore size distributions of membrane were characterized by a capillary flow porometer (POROLUXTM 100 FM, IB-FT, Germany). The base weight of mat was acquired by an electronic balance (MS105DU, Mettler-Toledo Group, Switzerland). The thickness of fibrous networks was gauged from their cross-sectional SEM images by the Nano Measurer software. The visible light transmittance of specimens was measured through a UV-Vis-NIR spectrometer (UV3600, Shimadzu Instruments Co., Ltd, Japan, with a standard barium sulfate whiteboard for the calibration) with a diffuse integrating sphere. Based on the USA standard (IEST-RP-CC52.2-2007) and the European standard (EN779: 2012) for air filters, the filtration performances of samples with effective testing area of 100 cm$^2$ were measured under continuous airflow velocity of

32 L min$^{-1}$ by an automated filtration testing machine (Model 8130, TSI Group, America). The instrument could produce charge-neutralized monodisperse solid NaCl particles with a mass median diameter of 260 nm and count median diameter of 75 nm. Each sample was tested for three times at different areas and then averaged to obtain final filtration results. The filtration efficiency (η) was automatically calculated by the commercial machine according to the equation (η = $1 - \frac{C}{C_0}$), where C and $C_0$ respectively represent the aerosol particle number concentrations at the testing outlet and the testing inlet. Quality factor (QF, Pa$^{-1}$) was calculated by the equation (QF = $-\frac{\ln(1-\eta)}{\Delta P}$), where ΔP is the pressure drop. The porosity (P) was calculated by the equation (P = $1 - \frac{w}{d\rho}$), where w, d and ρ respectively represent the base weight, thickness and density of the membrane. The long-term filtration process of the filter was conducted by the self-made filtration apparatus (continuous particle airflow containing over 100000 particles, Supplementary Fig. 7) and its filtration properties was then tested by commercial instrument. The surface potential of membrane was conducted by an electrostatic field tester (EFM 023, Kleinwaechter GmbH, Germany). The water vapor transmittance rate of filters was performed with a fabric moisture permeability meter (FX3081-CM15, TEXTEST, Switzerland) according to the standard of GB/T 12704.1–2009. The air permeability test was determined by a digital fabric air permeability meter (YG461E, Ningbo Textile Instrument Factory, China) according to the standard of GB/T 5453-1997. The infrared transmittance was obtained by the Fourier transform infrared (FTIR) spectrometer FTIR, Nicolet 8700, Thermo Fisher, USA) equipped with a diffuse gold integrating sphere (PIKE Technologies). The thermal images of filters worn by a volunteer (male, 28 years old) were recorded by an infrared camera (HM-TPK20-3AQF/W, Hangzhou Hikvision Electronics Co., Ltd., China). Based on the Chinese standard (GB/T 20944.3-2008), colony-counting method was utilized to quantitatively evaluate the antibacterial activity of the prepared filter[57]. The pure PA6 submicron-fiber membrane and PA6/DTAC dual-scale fibrous membrane were separately chosen as blank and experimental samples. And their antibacterial durability was also evaluated after six months of storage. The antibacterial rate (α) could be calculated by the equation (α = $-\frac{c_1}{c_2}$), where $c_1$ and $c_2$ represent the live bacterial concentration of blank and experimental samples. The live bacterial concentration is the product of colony count and dilution times of the bacterial culture solution. Error bars represent the standard deviation of the measured properties and are calculated from at least three samples.

## Reporting summary

Further information on research design is available in the Nature Portfolio Reporting Summary linked to this article.

## Data availability

The data that supports the findings of the study are included in the main text and supplementary information files. Source data are provided with this paper.

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

## Acknowledgements

This work was partly supported by the National Key Research and Development Program of China (2022YFB3804905 and 2022YFB3804900), the National Natural Science Foundation of China (52202218, 22075046, 22375047, 22378068 and 22378071), the Fundamental Research Funds for the Central Universities (2232020A-08), the Chang Jiang Scholars Program and the Innovation Program of Shanghai Municipal Education Commission (2019-01-07-00-03-E00023), Natural Science Foundation of Fujian Province (2022J01568).

## Author contributions

Y. Yang, X. Qin, and Y. Lai conceived and designed the research. Y. Yang carried out the experiments, analyzed data and wrote the manuscript. X. Li, Z. Zhou, Q. Qiu, and W. Cheng assisted in the characterization. J. Huang, W. Cai, X. Qin, and Y. Lai supervised the project and modified the manuscript. All authors discussed the results and reviewed the manuscript.

## Competing interests

The authors declare no competing interests.
