## [Peer Review File · Nature Communications]

Ultrathin, ultralight dual-scale fibrous networks with high-infrared transmittance for high-performance, comfortable and sustainable PM_{0.3} filterREVIEWER COMMENTS

Reviewer #1 (Remarks to the Author):

In this manuscript, the authors propose a facile yet massive method of antibacterial electrolyte-triggered splitting of charged jets to controllably prepare the high-performance, comfortable and sustainable fibrous filter. The designed hierarchically dual-scale fibrous membrane consists of continuous nanofibers (44 ± 7 nm) and submicron-fibers (159 ± 32 nm), which reduces the fiber diameter to a new level. As a result, the optimized filter shows ultralow thickness and base weight but superior protective ability of effective PM0.3 removal and durable antibacterial ability and wearing comfort of low air resistance, high heat dissipation and moisture permeability. Notably, ultralight filter can save more than 97% raw materials when compared to commercial face masks. The data is sufficient and the arguments are reasonable. The strategy and results are interesting and shed new insight for designing high-protective, comfortable, sustainable and low-cost personal protective materials. Thus, I recommend this work to be published in our journal after minor revision. The questions need to be addressed as follows:

1. Please explain why the microbeads in PA6 submicron fibers gradually decrease as the solution concentration increases from 3wt% to 5wt% (Fig. S1a-S1c).
2. The weight of the dual-scale fiber mask prepared in this paper is compared with that of the commercial N95 mask. The commercial N95 mask has not been verified for weight, so it is recommended to add the optical image for comparison (Fig.1d).
3. There is an error in the letter marking in the note of Fig. 1, and d is marked as c. Please modify it and check the details of the full text.
4. Please explain the phenomenon that PA6 filter shows strong electrostatic force but PA6/DTAC filter does not (Fig. 4g and 4h).
5. Fig. 5b and 5c compared the water vapor transmittance and air transmittance of Dual-scale fibers filter with MM and MBM, but did not mention the specific values of the three, so it is impossible to prove how many times the difference between the three. It is suggested to mark the specific values in Fig. 5b and 5c.
6. Please provide detailed methods for calculating porosity and quality factor in the experimental section.
7. The prepared dual-scale fibrous filter shows high visible light transmittance. Whether it has application prospects in anti-haze window screens.

Reviewer #2 (Remarks to the Author):

Electrospinning is not new and the work does not show any significant findings.

The paper lacks details. For example, how was the filtration efficiency determined, at what flow rate or face velocity, and using what instrument?

Reviewer #3 (Remarks to the Author):

Comments: This study described a facile and massive methodology to controllably fabricate the high-performance, comfortable, antibacterial and sustainable filter with high infrared transmittance via the one-step electrospinning technique. Although there are something new in this report, I think the manuscript should be revised seriously before publication. Here are some specific points which should be considered:

1. Efficiency and air resistance are the two most important factors of air filters. I suggest the authors should make a table or a plot to compare their results with the results calculated according to the literatures. Actually, the air resistance is not very low compared with the updated report.
2. Ultrathin, ultralight fiber filter contributes to its wearing comfort and sustainability of raw materials. However, ultrathin and ultralight fibers can also lead to reduced filtration capacity and service life of the filter. The filtration capacity or service life of the filter are not mentioned in this paper. The authors should add and discuss the results.
3. The images in Figure 4, such as figure 4d-f cannot clearly show the results. They should be improved.
4. Line 377-379. "The efficient and long-term antibacterial properties of resultant filter could be attributed to the nano- and submicron-scale fiber size and sustained release of DTAC from fiber inside to surface." DTAC also irritates human skin and seriously irritates eyes. The authors need to evaluate the safety of the filter.
5. The antibacterial activity of a material is usually judged by the size of the antibacterial zone of the material. By what method were the conclusions in Figure 5g tested? How to get the results?

Reviewer #1 (Remarks to the Author):

In this manuscript, the authors propose a facile yet massive method of antibacterial electrolyte-triggered splitting of charged jets to controllably prepare the high-performance, comfortable and sustainable fibrous filter. The designed hierarchically dual-scale fibrous membrane consists of continuous nanofibers (44 ± 7 nm) and submicron-fibers (159 ± 32 nm), which reduces the fiber diameter to a new level. As a result, the optimized filter shows ultralow thickness and base weight but superior protective ability of effective $PM_{0.3}$ removal and durable antibacterial ability and wearing comfort of low air resistance, high heat dissipation and moisture permeability. Notably, ultralight filter can save more than 97% raw materials when compared to commercial face masks. The data is sufficient and the arguments are reasonable. The strategy and results are interesting and shed new insight for designing high-protective, comfortable, sustainable and low-cost personal protective materials. Thus, I recommend this work to be published in our journal after minor revision. The questions need to be addressed as follows:

Answer: We appreciate your affirmation of our research and valuable comments. We have carefully revised our manuscript point-by-point according to the comments.

Q1: Please explain why the microbeads in PA6 submicron fibers gradually decrease as the solution concentration increases from 3wt% to 5wt% (Fig. S1a-S1c).

A1: Thanks for your comment. The formation of eletrospun beaded fibers results from the Rayleigh instability of polymeric jets.^[1] Dilute jets from low viscosity solution exhibits poor chain entanglement, which results in their uneven stretching (Rayleigh instability) and form into contracted beads.^[1] The increased solution concentration enhances its chain entanglement and viscosity, which enable jets to effectively resist the Rayleigh instability and form into non-beaded fibers. Above explanations have also been added in the revised manuscript.

[1] J. Xue, T. Wu, Y. Dai, Y. Xia. Electrospinning and Electrospun Nanofibers: Methods, Materials, and Applications. Chem. Rev. 119(8), 5298-5415 (2019).

Q2: The weight of the dual-scale fiber mask prepared in this paper is compared with that of the commercial N95 mask. The commercial N95 mask has not been verified for weight, so it is recommended to add the optical image for comparison (Fig.1d).

A2: Thank you for the helpful suggestion. Considering the limited space of Fig. 1d, the optical image for verifying N95 mask weight has been added in the Fig. S1.

Figure S1. The optical image for verifying the weight of commercial N95 mask.

Q3: There is an error in the letter marking in the note of Fig. 1, and d is marked as c. Please modify it and check the details of the full text.

A3: Thanks a lot for pointing out the mistake. It has been corrected in the revised manuscript and we also carefully reviewed the manuscript twice again.

Q4: Please explain the phenomenon that PA6 filter shows strong electrostatic force but PA6/DTAC filter does not (Fig. 4g and 4h).

A4: Thank you very much for the valuable comment. Electrospinning is an in-situ charging process and thus endowed hydrophobic PA6 filter with strong electrostatic force. Differently, as an ionic electrolyte, DTAC with high conductivity could quickly dissipate the generated electrostatic charges during electrospinning process. Moreover, hydrophilic PA6/DTAC fibers (Fig. 3a) could also combine with water molecules and thus neutralize the electrostatic charges. As a result, PA6/DTAC filter did not exhibit electrostatic effect.

Q5: Fig. 5b and 5c compared the water vapor transmittance and air transmittance of Dual-scale fibers filter with MM and MBM, but did not mention the specific values of the three, so it is impossible to prove how many times the difference between the three. It is suggested to mark the specific values in Fig. 5b and 5c.

A5: Thanks a lot for your kind suggestions. The specific values of the water vapor transmittance and air transmittance of the three filters have been marked in the revised manuscript.

Q6: Please provide detailed methods for calculating porosity and quality factor in the experimental section.

A6: Thank you for the valuable comments. The porosity (P) was calculated by the equation ($P=1-w/d\rho$), where w , d and ρ respectively represent the base weight, thickness and density of the membrane. Quality factor (QF, Pa^{-1}) was calculated by the equation ($QF=-\ln(1-\eta)/\Delta P$), where η and ΔP are the filtration efficiency and pressure drop. The calculation details have been added in the revised manuscript.

Q7: The prepared dual-scale fibrous filter shows high visible light transmittance. Whether it has application prospects in anti-haze window screens.

A7: Thank you very much for the significant advice. The dual-scale fibrous filter with high visible light transmittance is expected to be utilized as anti-haze window screens. However, integrating efficient PM removal, high visible light transmittance, and sufficient mechanical properties into dual scale fiber-based window screens still faces some practical challenges, which we will explore in depth in our future work.

Reviewer #2 (Remarks to the Author):

Q1: Electrospinning is not new and the work does not show any significant findings.

A1: We respectfully disagree with some assessments from the reviewer, and have carefully clarified the concerns that the reviewer raised (see our responses below). High-performance, comfortable and sustainable filters for personal protection have been calling for the ultrathin and ultralight fibrous membrane composed of true nanofibers (diameter <100 nm). As one of the most promising

techniques for fabricating ultrafine fibers, however, electrospinning can only produce fibers with pseudo-nanoscale diameter (>100 nm). In this manuscript, the novel antibacterial electrolyte-triggered splitting electrospinning strategy was developed to massively prepare dual-scale fibrous membrane consisting of continuous nanofibers (44 ± 7 nm) and submicron-fibers (159 ± 32 nm). These material and structural merits endow the ultrathin, ultralight filter with high-protective, comfortable, sustainable and low-cost characteristics. More importantly, the facile one-step splitting electrospinning method reduces the fiber diameter to a new level and may open a new insight for designing hierarchical nanofibrous membrane and exploiting advanced filtration and separation materials.

Q2: The paper lacks details. For example, how was the filtration efficiency determined, at what flow rate or face velocity, and using what instrument?

A2: Thank you very much for the kind suggestions. Based on the USA standard (IEST-RP-CC52.2-2007) and the European standard (EN779: 2012) for air filters, the filtration performances of samples with effective testing area of 100 cm^2 were measured under continuous airflow velocity of 32 L/min by an automated filtration testing machine (Model 8130, TSI Group, America). Above information was included in the experimental section of the manuscript.

Reviewer #3 (Remarks to the Author):

Comments: This study described a facile and massive methodology to controllably fabricate the high-performance, comfortable, antibacterial and sustainable filter with high infrared transmittance via the one-step electrospinning technique. Although there are something new in this report, I think the manuscript should be revised seriously before publication. Here are some specific points which should be considered:

Answer: Very grateful for your positive and constructive comments. According to your remarks, we have complemented the required experiments, provided indepth analysis and discussion, added experimental details and carefully revised our manuscript.

Q1: Efficiency and air resistance are the two most important factors of air filters. I suggest the authors should make a table or a plot to the compare their results with the results calculated according to the literatures. Actually, the air resistance is not very low compared with the updated report.

A1: Thank you very much for the meaningful suggestion. The comparison of filtration efficiency and air resistance between our work and recent works has been added as shown below. In comparison to these excellent works in recent years, the dual-scale fibrous filter we designed could integrate the ultrathin and ultralight characteristics and superior filtration properties, which enable itself to be high-protective, comfortable and sustainable. Of course, we are exploring to further improve the filtration efficiency, wearing comfort and reduce the filtration resistance by exploiting novel hierarchical nanofibrous filters. And the following table was also complemented in the Supporting formation (Table S1).

Table S1. Filtration performances comparison between our work and those from the recent literatures.

Supporting information Reference	Base weight (g/m ²)	Thickness (μm)	PM size (μm)	Airflow (L/min)	Filtration efficiency (%)	Pressure drop (Pa)
[1]	-	-	PM _{2.5}	180	99.59	26
[2]	10	-	PM _{0.3}	5	97	10
[3]	-	-	PM _{0.3}	32	99.97	189
[4]	0.64	-	PM _{0.3}	32	99.1	78
[5]	6	20	PM _{0.3}	210	98.1	84
[6]	31.2	10.16	PM _{1.0}	32	91	91
[7]	-	13	PM _{0.3}	32	99.2	90
[8]	-	-	PM _{0.3}	32	99.3	127.4
[9]	5.4	-	PM _{0.3}	32	96.1	57
[10]	80	517	PM _{0.3}	32	99.97	234
[11]	-	3.9	PM _{0.3}	32	95	53
Our work	0.24	-	PM _{0.3}	32	97.4	56
Our work	0.36	-	PM _{0.3}	32	99.5	72
Our work	0.57	1.49	PM _{0.3}	32	99.95	120
Our work	0.69	-	PM _{0.3}	32	99.98	130

Reference

- [1] G. Zhang, Q. Zhu, L. Zhang, F. Yong, Z. Zhang, S. Wang, Y. Wang, L. He, G. Tao. High-performance particulate matter including nanoscale particle removal by a self-powered air filter. *Nat. Commun.* 11, 1653 (2020).
- [2] Q. Wang, Y. Wei, W. Li, X. Luo, X. Zhang, J. Di, G. Wang, J. Yu. Polarity-dominated stable N97 respirators for airborne virus capture based on nanofibrous membranes. *Angew. Chem. Int. Ed.* 60, 23756-23762 (2021).
- [3] F. Wang, Y. Si, J. Yu, B. Ding. Tailoring Nanonets-engineered superflexible nanofibrous aerogels with hierarchical cage-like architecture enables renewable antimicrobial air filtration. *Adv. Funct. Mater.* 31, 2107223 (2021).
- [4] Z. Yang, X. Zhang, Z. Qin, H. Li, J. Wang, G. Zeng, C. Liu, J. Long, Y. Zhao, Y. Li, G. Yan. Airflow Synergistic Needleless Electrospinning of instant noodle-like curly nanofibrous membranes for high-efficiency air filtration. *Small* 18, 2107250 (2022).
- [5] Z. Peng, J. Shi, X. Xiao, Y. Hong, X. Li, W. Zhang, Y. Cheng, Z. Wang, W. J. Li, J. Chen, M. K. H. Leung, Z. Yang. Self-charging electrostatic face masks leveraging triboelectrification for

- prolonged air filtration. *Nat. Commun.* 13, 7835 (2022).
- [6] T. Le, E. Curry, T. Vinikoor, R. Das, Y. Liu, D. Sheets, K. Tran, C. Hawxhurst, J. Stevens, J. Hancock, O. Bilal, L. Shor, T. Nguyen. Piezoelectric nanofiber membrane for reusable, stable, and highly functional face mask filter with long-term biodegradability. *Adv. Funct. Mater.* 32, 2113040 (2022).
- [7] C. Wang, N. Meng, A. Babar, X. Gong, G. Liu, X. Wang, J. Yu, B. Ding. Highly transparent nanofibrous membranes used as transparent masks for efficient PM_{0.3} removal. *ACS Nano* 16, 119-128 (2022).
- [8] X. Gong, C. Jin, X. Liu, J. Yu, S. Zhang, B. Ding. Scalable fabrication of electrospun true-nanoscale fiber membranes for effective selective separation. *Nano Lett.* 23, 1044-1051 (2023).
- [9] S. Shi, Y. Si, Z. Li, S. Meng, S. Zhang, H. Wu, C. Zhi, W.-F. Io, Y. Ming, D. Wang, B. Fei, H. Huang, J. Hao, J. Hu. An intelligent wearable filtration system for health management. *ACS Nano* 17, 7035-7046 (2023).
- [10] Z. Zhou, T. You, D. Wang, Z. Pan, G. Xu, Y. Liang, M. Tang. Conformal build-up of functionalized air filters with improved air cleaning and bioprotective traps. *Adv. Funct. Mater.* 33, 2306777 (2023).
- [11] C. Wang, X. Wang, J. Yu, B. Ding. Highly transparent carbon nanofibrous membranes inspired by dragonfly wings. *ACS Nano* 17, 10888-10897 (2023).

Q2: Ultrathin, ultralight fiber filter contributes to its wearing comfort and sustainability of raw materials. However, ultrathin and ultralight fibers can also lead to reduced filtration capacity and service life of the filter. The filtration capacity or service life of the filter are not mentioned in this paper. The authors should add and discuss the results.

A2: Thanks a lot for your beneficial suggestion. As suggested, the service life of the prepared filter has been evaluated by the 50 filtration cycles and 3 hours of long-term filtration test by commercial machine. The long-term filtration process of the filter was conducted by the self-made filtration apparatus (continuous particle airflow containing over 100000 particles, Supplementary Fig. 7) and its filtration properties was then tested by commercial instrument. After 50 filtration cycles, its PM_{0.3} filtration efficiency and filtration resistance respectively increased from 99.958% to 99.986% and 119 Pa to 126 Pa (Fig. 4g). Similarly, after 3 hours of long-term filtration test, the PM_{0.3} filtration efficiency and filtration resistance of the dual-scale fibrous filter also increased from 99.961% to 99.986% and 119 Pa to 128 Pa (Fig. 4h). The results indicated that the designed filter showed superior filtration durability and service life. The increase in both filtration efficiency and pressure drop was mainly due to pore clogging caused by the accumulation of specific substances in porous fibrous filter after persistent filtration. Above results and discussion have been added in the revised manuscript.

Figure 4. (g) Filtration cycle and (h) long-term filtration of the dual-scale fibrous filter.

Figure S7. Optical images of self-made apparatus for long-term filtration.

Q3: The images in Figure 4, such as figure 4d-f cannot clearly show the results. They should be improved.

A3: Thanks a lot for your kind advice. Figure 4d-f has been modified as shown below.

Figure 4. (d) Filtration properties of PA6 and PA6/DTAC-2 filters. Filtration performances of PA6/DTAC-2 filter (e) placed in various relative humidity and (f) under different airflow rate.

Q4: Line 377-379. “The efficient and long-term antibacterial properties of resultant filter could be attributed to the nano- and submicron-scale fiber size and sustained release of DTAC from fiber inside to surface.” DTAC also irritates human skin and seriously irritates eyes. The authors need to evaluate the safety of the filter.

A4: Thank you very much for the constructive comment. The main purpose of choosing DTAC was to induce jet splitting for preparing dual-scale fibrous membrane composed of true nanofibers and submicron-fibers. And DTAC accounted for only 3wt% of PA6 dual-scale fibrous membrane. Besides, the filter with sandwich structure could also isolate DTAC-doped PA6 fibers from human skin and eyes to some extent. What's more, our research group is developing one kind of more human-friendly antibacterial electrolyte, thereby facilitating the promising application of the designed dual-scale fibrous air filters.

Q5: The antibacterial activity of a material is usually judged by the size of the antibacterial zone of the material. By what method were the conclusions in Figure 5g tested? How to get the results?

A5: Thanks a lot for your helpful comment. The size of the antibacterial zone of the material (inhibition zone method) can only qualitatively judge its antimicrobial activity. In this work, based on the Chinese standard (GB/T 20944.3-2008), colony-counting method was utilized to quantitatively evaluate the antibacterial activity of the prepared filter. The pure PA6 submicron-fiber membrane and PA6-DTAC dual-scale fibrous membrane were separately chosen as blank and experimental samples (Fig. 5g). And their antibacterial durability was also evaluated after six months of storage. The antibacterial rate (α) could be calculated by the equation ($\alpha=1-c_1/c_2$), where c_1 and c_2 represent the live bacterial concentration of blank and experimental samples. The live bacterial concentration is the product of colony count and dilution times of the bacterial culture solution. Above details have been added in the revised manuscript.

REVIEWERS' COMMENTS

Reviewer #1 (Remarks to the Author):

Dear author:

Thank you very much for answering my doubts. This manuscript is now suitable for publication.

Reviewer #3 (Remarks to the Author):

Accept.

Reviewer #1 (Remarks to the Author):

Dear author:

Thank you very much for answering my doubts. This manuscript is now suitable for publication.

Answer: We appreciate your affirmation of our research and constructive suggestions for improving our manuscript.

Reviewer #3 (Remarks to the Author):

Accept.

Answer: We sincerely thank the reviewer for the valuable comments and the recommendation of our paper to be published in *Nature Communications*.